# pH-Related Changes in Soil Bacterial Communities in the Sanjiang Plain, Northeast China

**DOI:** 10.3390/microorganisms11122950

**Published:** 2023-12-09

**Authors:** Mingyu Wang, Wenmiao Pu, Shenzheng Wang, Xiannan Zeng, Xin Sui, Xin Wang

**Affiliations:** 1Engineering Research Center of Agricultural Microbiology Technology, Ministry of Education & Heilongjiang Provincial Key Laboratory of Ecological Restoration and Resource Utilization for Cold Region & Key Laboratory of Microbiology, College of Heilongjiang Province & School of Life Sciences, Heilongjiang University, Harbin 150080, China; wmy022234@163.com (M.W.); puwenmiao@163.com (W.P.); wangshenzheng2000@163.com (S.W.); 2Institute of Crop Cultivation and Tillage, Heilongjiang Academy of Agricultural Sciences, Harbin 150088, China; zengxiannanzxn@163.com

**Keywords:** soil bacteria, alpha diversity, soil pH, nitrogen, organic matter

## Abstract

Soil bacteria are crucial components of terrestrial ecosystems, playing an important role in soil biogeochemical cycles. Although bacterial community diversity and composition are regulated by many abiotic and biotic factors, how soil physiochemical properties impact the soil bacteria community diversity and composition in wetland ecosystems remains largely unknown. In this study, we used high-throughput sequencing technology to investigate the diversity and composition of a soil bacterial community, as well as used the structural equation modeling (SEM) method to investigate the relationships of the soil’s physicochemical properties (i.e., soil pH, soil organic carbon (SOC), total nitrogen (TN), ammonium nitrogen (NH_4_^+^N), electrical conductivity (EC) and nitrate nitrogen (NO_3_^−^N)), and soil bacterial community structures in three typical wetland sites in the Sanjiang Plain wetland. Our results showed that the soil physicochemical properties significantly changed the α and β-diversity of the soil bacteria communities, e.g., soil TN, NH_4_^+^N, NO_3_^−^N, and SOC were the main soil factors affecting the soil bacterial α-diversity. The soil TN and pH were the key soil factors affecting the soil bacterial community. Our results suggest that changes in soil pH indirectly affect soil bacterial communities by altering the soil nitrogenous nutrient content.

## 1. Introduction

Wetlands have the unique characteristics of land and water systems, with distinct ecosystem structures, processes, and functions, e.g., water purification, water conservation, flood storage and regulation, pollution degradation, climate regulation, and regional water cycles [1,2]. Therefore, wetland ecosystems provide important contributions to ecotourism and biodiversity conservation [3,4,5]. Soils are the dominant component of wetlands and play important ecological functions in the wetland biogeochemical cycle and vegetation composition [6]. Soil microorganisms, as the dominant life component in soil, regulate many ecological functions, e.g., soil formation and development, soil carbon and nitrogen cycles, and vegetation succession [7,8,9]. Therefore, research on soil microbiomes under different ecosystem types and driving mechanisms are popular topics in current ecological studies [10,11,12].

Bacteria, as an important component of soil in wetland ecosystems, represent the majority of the biodiversity of terrestrial ecosystems and are closely associated with the cycling of various elements, such as in the carbon and nitrogen cycles [13,14]. However, the soil’s microbial community diversity and composition varies widely between different ecosystems due to the variation characteristics of biotic and abiotic factors [15,16,17]. Soil bacteria are exposed to an increasing number of environmental disturbances as human activities continue to intensify. For example, some studies have investigated microbial community assembly in wetlands at local scales [18,19,20,21]. Furthermore, the changes in the structures, functions, and diversities of soil microbial communities are closely related to the variations in soil properties [22]. Therefore, understanding the microbial community structure and the key environmental factors that drive the composition and diversity of wetland soil microbial communities is important for disclosing the relationships between soil microorganisms and wetland ecosystem function.

The Sanjiang Plain is the largest fresh wetland distribution region in the northeast of China, with abundant natural resources. It plays indispensable roles in maintaining the regional balance of nutrient cycling, and global climate stability, serving as an important ecological barrier for the northeast of China [23,24]. Currently, this region is a research hot spot regarding global changes and human activities. The researches on the above-ground vegetation community distribution, landscape pattern, and hydrological resources of the Sanjiang Plain have been extensively studied [25,26,27,28,29]. These results indicate that the biomass, species composition, and diversity of vegetation in the Sanjiang Plain changed significantly in response to various human activities and geographic spaces [25,26,27]. Soil microorganisms are closely related to geographic space and vegetation. We, therefore, assume that the soil microbiome may change due to the changes in the aboveground vegetation in different geographic regions. For instance, Sui et al. (2019) reported that land use patterns significantly affected the α-diversity of the soil microbiome in the Sanjiang Plain [30,31,32,33]. Moreover, previous studies have indicated that research on the driving environmental factors on the soil microbial community did not allow for a unified conclusion [34,35,36]. For example, Shu et al. [37] identified soil pH as the most critical factor shaping bacterial community composition, but Cao et al. [38] emphasized the importance of soil total phosphorus in shaping soil bacterial community diversity in Sanjiang plain. Therefore, the research on the structure and diversity of the soil microbial communities in the Sanjiang Plain wetland is full of controversy.

In this study, we selected three typical wetland ecosystems located in Qixinghe, Naolihe, and Honghe national nature reserves, and investigated the soil bacterial community structure and diversity and its relationships with the soil’s physicochemical properties by using high-throughput sequencing technology. The main aims of this study were (a) to compare and analyze the variability of the soil bacterial communities in these three nature reserves belonging to the Sanjiang Plain; and (b) to clarify which soil properties are the main drivers of the soil bacterial community structure. This study is important for understanding the soil quality and nutrient conversion in the context of global climate change, as well as providing scientific data for exploring changes in soil microbial communities in the wetlands of the Sanjiang Plain.

## 2. Materials and Methods

### 2.1. Study Area

This study is located in the Sanjiang Plain wetland in the eastern region of Heilongjiang Province (45°01′–48°27′56″ N, 130°13′–135°05′26″ E) of China (Figure 1). The region features a humid, semi-humid continental monsoon climate with an average annual temperature ranging from 1 to 4 °C. The annual temperature ranges from 2300 °C to 2500 °C in accumulation. The study was performed in June when the soil temperature varied from 21 °C to 22 °C. The average altitude of the area is between 50 and 60 m above sea level. Predominant soil types are Haplic Phaeozems. The dominant vegetation primarily consists of *D. Angustifolia*, *S. Rosmarinifolia*, *Carex lasiocarpa*, and *Phragmites australis*.

### 2.2. Soil Sampling

Soil samples were collected in June 2022 at three nature reserves (Honghe National Nature Reserve (NW) (47°42′18″–47°52′07″ N, 133°34′38″–133°46′29″ E), Naoli River National Nature Reserve (NLH)(46°30′22″–47°24′32″ N, 132°22′29″–134°13′45″ E) and Qixing River National Nature Reserve (QXH)(46°40′–46°52′ N, 132°05′–132°26′ E). In each nature reserve, three sampling plots (*n* = 3) were randomly selected (total 9 plots). Within each plot, soil samples (0–20 cm) were collected from 10–15 soils along an S-shaped path using an 8 cm diameter soil auger, and all soil samples obtained were mixed after removal of surface debris (e.g., leaves and dry vegetation) to ensure that soil samples from each plot were representative. The soil samples were then placed in sterilized ziplock bags and stored at −20 °C in a refrigerator. The soil samples were transferred to the laboratory immediately and sieved (2 mm screen) to remove gravel and plant material at the laboratory. The soil samples were divided into two parts; one was air-dried for physicochemical analyses, and the other was stored at −80 °C for subsequent microbiological analyses.

### 2.3. Characterization of Soil Physicochemical Parameters

The soil temperature was tested using a soil thermometer, with temperature differences between sites during sampling ranging from 1–2 °C. We used the gravimetry method to measure the soil moisture content. Soil pH was measured using a pH meter (Delta 320, Mettler Toledo, Greifensee, Switzerland) with a 1:2.5 (*w*/*v*) mixture of soil and deionized water [39]. Soil EC was measured by a conductivity meter (Multiparameter SevenExcellence™) [40]. The concentration of nitrate nitrogen (NO_3_^−^N) in soils was determined by extraction with KCl and measurement with a continuous flow analyzer (AutoAnalyzer 3, SEAL, Darmstadt, Germany) [41]. Total phosphorus (TP) was determined by NaOH fusion-molybdenum-antimony colorimetric method, and available phosphorus was determined by the NaHCO_3_ molybdenum colorimetric method [42]. Soil total nitrogen (TN) was determined by an elemental analyzer (VarioEL III, Langenselbold, Germany) [43,44]; Soil ammonium nitrogen (NH_4_^+^N) content was measured using a continuous flow analysis system (SKALAR SAN++, Breda, The Netherlands) [45]. Soil organic carbon (SOC) was determined using an elemental analyzer (VarioEL III, Langenselbold, Germany) [46].

### 2.4. DNA Extraction, PCR Amplification, and MiSeq Sequencing

DNA extraction was performed using a PowerSoil kit to extract DNA from approximately 500 mg of soil [47,48]. The quantity and quality of DNA were first detected by agarose gel electrophoresis (1%) and then determined by NanoDrop ND-1000 spectrophotometer (Thermo Fisher Scientific; Waltham, MA, USA). After DNA extraction, the relative abundance of the 16S rRNA gene was detected by real-time fluorescence quantitative PCR (qPCR) [49]. For each soil sample, the qPCR reaction was repeated four times. The 16S rRNA gene was amplified using primers 341F (5′-CCTACGGGNGGCWGCAG-3′) and 805R (5′-GACTACHVGGTATCTAATCC-3′) [50,51]. The target gene amplification conditions were 30 cycles of initial denaturation for 5 min at 94 °C, 30 s at 94 °C, 30 s at 55 °C, 50 s at 72 °C, and a final extension of 5 min at 72 °C. Repeated PCR reactions were carried out for each sample, and the polymerized PCR products were subjected to 1% agarose gel electrophoresis. Bands of the correct size were taken and PCR products were purified using the TAKARA DNA gel extraction kit. All PCR products were quantified by Nanodrop. Sequencing samples were performed using the TruSeq DNA kit. The purified libraries were diluted, denatured, re-diluted according to the Illumina library preparation protocol, mixed with 30% of the final DNA amount of PhiX, and finally sequenced. PCR products were pooled in equal proportions and sequenced using an Illumina MiSeq PE300 sequencer [52,53,54].

### 2.5. Analysis of Sequencing Data

The raw data was analyzed using the QIIME1 and UPARSE pipelines. Overall, UPARSE was used for classification assignments with a similarity threshold greater than 97%. We used the SILVA database (version 132) to classify the bacteria taxonomy. Operational taxonomic identities were determined using QIIME1 by performing the BLAST algorithm on sequences in the SILVA database (version 132). The abundance of each operational taxonomic unit (OTU) for each sample and the taxonomy of these OTUs were then tabulated. α-diversity indices [55] (e.g., ACE index, Chao1 index, Simpson index, Shannon index) in QIIME1 were calculated using the OTU table described above.

### 2.6. Statistical Analyses

In R v4.3.1, Venn diagrams were generated using the “Venn Diagram” package to visualize unique and shared OTUs in the samples [56,57], and the package allows the visualization of unique and shared OTUs in the samples. Similarly, heatmaps of the top 20 genera in each soil sample were generated in R using the “ggplot2” and “pheatmap” packages. One-way analysis of variance (ANOVA) and least significant difference (LSD) analyses were used to determine the effects of treatments on different microbial communities in different wetland types. Cladogram diagrams illustrating taxonomic relationships from phylum to species were generated on the online platform (https://www.bioincloud.tech/standalone-task-ui/lefse (accessed on 1 November 2023)). PCA analyses were performed using the “Vegan” package to determine if there were significant differences in soil bacterial communities between groups. One-way analysis of variance (ANOVA) and least significant difference tests (LSD) were used to determine changes in soil physicochemical parameters, changes in bacterial alpha diversity, and relative abundance of different bacterial communities in different wetland types at 0.05 levels [58,59], by using the SPSS version 24.0 (Chicago, IL, USA) [60]. We performed a correlation analysis (Pearson correlation) using the “ggcor” package to clarify the correlation between bacterial α-diversity and soil properties [61]. Structural equation modeling (SEM) was used to detect the direct and indirect effects of soil properties on the diversity and composition of soil bacterial communities [62,63]. We constructed an a priori model based on the literature review and our knowledge of the relationships between these predictors. Model adequacy was determined by χ^2^ tests (*p* > 0.05) [64], comparative fit index (CFI, >0.9), Akaike Information Criteria (AIC), Bayesian Information Criteria (BIC) and Standardized Root Mean Square Residual (SRMR, <0.08).

## 3. Results

### 3.1. Soil Physicochemical Characteristics

All soil physicochemical parameters differed significantly among the three nature reserves (Table 1, *p* < 0.05). Soil pH (7.3) and SWC (45.3%) in QXH were significantly higher than the other two nature reserves (*p* < 0.05) (Table 1). The contents of other soil physicochemical parameters in QXH were significantly lower than those in NW and NLH (Table 1, *p* < 0.05).

### 3.2. Effect of Different Wetland Types on Soil Bacterial Diversity

The α-diversity of the soil bacterial community differed significantly among the three nature reserves (Table 2, *p* < 0.05), except for Shannon index. The α-diversity of the soil bacterial community of NW nature reserve exhibited the highest OTUs, ACE and Chao1 indices than those of the other two nature reserves, whereas the α-diversity of the soil bacterial community of QXH nature reserve had the lowest OTU observed number, ACE index, and Chao1 indices than those of other two nature reserves (Table 2). However, the Shannon index did not change significantly among the three nature reserves (Table 2).

### 3.3. Effects of Different Nature Reserves on Bacterial Community Structure

Most of the top 20 most abundant genera were Gram-negative, with MND1 and RB41 being the two most abundant genera only in the QXH, while the genera Haliangium and ADurb were the two most abundant genera in the NLH nature reserve (Figure 2). The structures of the bacterial communities, particularly the unique and shared OTUs among three nature reserves, were visualized using Venn diagrams (Figure 3). A total of 206 OTUs were shared between three wetland types (Figure 3). Addition, 10 OTUs only existed in NLH, 2 OTUs only existed in NW nature reserve, 2 OTUs only existed in QXH nature reserve (Figure 3). The PCA demonstrated that the soil bacterial community structures were significantly different between each nature reserve; the two axis of PCA explained 45.7% of the variance (Figure 4).

### 3.4. Indicator Species

Significant differences were observed among 16 taxa in the three nature reserves, as indicated by LDA effect size scores of >4.0 (Figure 5).

### 3.5. Relationship of Bacterial Community and Soil Physicochemical Characteristics

The observed number, ACE index, and Chao1 index were significantly and positively correlated with contents of soil SOC, TN, TP, AP, NO_3_^−^N, NH_4_^+^N and EC (Figure 6), whereas the observed number, ACE index, and Chao 1 index were significantly and negatively correlated with soil pH and SWC (Figure 6). Simpson index was significantly and positively correlated with soil pH (Figure 6, *p* < 0.05) (Figure 6). SEM was used to assess the direct and indirect effects of soil properties on soil bacterial α-diversity and composition. The final model significantly fit criteria (χ^2^ = 18.545, df = 14, *p* = 0.183, SRMR = 0.038, CFI = 0.976, AIC = 149.24, BIC = 153.38) (Figure 7). Soil TN indirectly affected the α-diversity of soil bacterial community, while NH_4_^+^N, NO_3_^−^N, and SOC had a significant direct effect on α-diversity (Figure 7). TN and pH significantly directly affected the composition of the soil bacterial community (Figure 7).

## 4. Discussion

### 4.1. Differences of Soil Bacterial Communities in Three Different Wetlands

Soil microorganisms are vital for nutrient cycling, organic matter decomposition, soil structure maintenance and plant growth in ecosystems [65,66,67]. Soil microbial diversity, encompassing genetic, species, and ecosystem-level variations serves as an indicator of microbial community stability and is assessed using α-diversity, which combines species richness and evenness [68,69]. In this study, all soil physicochemical parameters differed significantly among the three nature reserves. Specifically, soil pH and soil water content (SWC) in QXH were notably higher than the other two nature reserves (*p* < 0.05). Conversely, the contents of other soil physicochemical parameters in QXH were considerably lower than those in NW and NLH (Table 1). We also found that the soil bacterial α-diversity showed great variation between the three nature reserves. The NW nature reserve had the highest richness indices (OTU observations, ACE and Chao1 indices), but the QXH nature reserve had the lowest richness indices. Shannon index of soil bacterial community did not change among the three nature reserves. The Shannon index is one of the indices to estimate the diversity of soil microbiome. A higher Shannon index indicates higher species diversity. The ACE index and the Chao1 index represent the species richness and are sensitive to rare species, therefore, we supposed that the NLH and NW nature reserves might have the more rare species than NW nature reserve [70,71]. Our results found that the species richness in NW nature reserve was higher, but lower in QXH nature reserve. However, there was no difference in species diversity between the three reserves.

This study found that soil pH was the main driving factor in the variations of soil bacterial community diversity, which is consistent with Pan et al. [72]. However, some studies discovered that EC was the key factor determining the composition of bacterial communities [73]. This discrepancy is possibly related to the soil types in different ecosystems, such as soils in saline–alkali areas being significantly influenced by EC [73]. Enzyme catalysis is the main process for the growth of soil microorganisms and is regulated by soil pH [74]. Yang et al. (2021) reported that soil pH was the main driver of changes in soil microbial diversity by conducting a global meta-analysis [75]. We supposed that pH can also modulate the permeability of microbial cell membranes and the solubility or ionization of substances, thereby exerting an influence on nutrient absorption, consequently affecting microbial growth and development. Additionally, some studies have demonstrated that soil nutrient content also affected the diversity and structure of soil bacterial communities [76,77]. This study found that the relative abundance of soil bacterial phyla or genera was significantly positively correlated with soil nutrient content, but soil bacterial diversity did not correlate with soil nutrient content (Figure 6). This indicated that the changes of soil nutrient content did not affect soil bacterial diversity, but only prompted the growth of some soil bacterial phyla or genera. This is consistent with many previous studies; for example, Yao et al. [78] found that the addition of some nutrients increased the richness of soil bacteria in cultivated fields. However, inconsistent with our result, Yao et al. [78] also found that the diversity of soil bacteria was suppressed, this may be due to the difference in the experimental geography and wetland ecosystems were a wet ecosystems compared to other ecosystems, and soil nutrient addition would be diluted to a large extent. This can explain why these three nature reserves with a higher nutrient content had higher soil bacterial abundance in our study.

### 4.2. The Main Drivers of Bacteria Community in the Soil Characteristic

The structure of soil microbial communities can provide insights into the abundance, diversity, and stability of the microbial population, playing a crucial role in keeping soil quality and ecosystem productivity [79,80,81]. In this study, we found that the most 20 abundant genera were Gram-negative. The research region belonged to seasonal waterlog, and the soil samples were conducted during a sunny period when no waterlogging and well-aerated conditions occurred in the research sites., We, therefore, assume that well-aerated soil conditions contributed to the competitive advantage gained by Gram-negative bacteria. This is consistent with previous studies, such as Balasooriya et al. (2008) found that Gram-negative bacteria preferred aerobic conditions and that soil aeration was a key determinant of Gram-negative bacterial communities [82]. We also observed that the relative abundance of soil bacterial genera *MND1* and *RD41* spp. were much higher in the QXH nature reserve, but were rarer in the other two nature reserves. Notably, *Streptomyces* was not the dominant genera in this study; this is inconsistent with other studies [83,84]. This is possibly due to the fact that *Streptomyces* usually grow in alkaline soils, but the soil samples in the Sanjiang plain wetland were acidic [84] (Figure 2). In total, 206 OTUs were shared in three wetland nature reserves, and 10 endemic OTUs were present only in NW Nature Reserve, 2 endemic OTUs were only present in NLH, and 2 endemic OTUs were in QXH nature reserve (Figure 3). The principal component analysis (PCA) indicated the soil bacterial communities were significantly different in the three nature reserves (Figure 4). The Cladogram analysis revealed that the *Haliangium* and *ADurb.Bin063-1* were indicative of genera for the NLH nature reserve, and *MND1* was indicative of genera in the QXH nature reserve (Figure 5). Previous studies have demonstrated that soil physicochemical properties were the main factors affecting the soil bacterial community composition [85,86]. Previous studies reported that *MND1* played a pivotal role in soil nitrogen cycling [86]. *RB41* exhibited the remarkable ability to selectively regulate the decomposition of soil organic matter and the carbon cycling process in agricultural soils and positively correlated with soil pH [87,88,89]. We believe that this is the reason that certain bacteria have evolved effective strategies to thrive in extreme environments. Several studies have demonstrated the adaptation of extreme microorganisms, such as high salt adaptation in halophiles and metal adaptation in acidophiles [90], which further supports our perspective. In general, non-biotic factors play a crucial role in shaping the composition of soil microbial communities. Numerous studies showed that soil organic carbon (SOC), soil pH, total nitrogen, and other soil physicochemical properties may impact the composition of soil microbial communities. Additionally, interactions among two or more soil physicochemical properties may lead to synergistic or antagonistic effects.

Analyzing only the correlations between soil physicochemical properties and the composition of soil bacteria cannot fully elucidate the primary influencing factors on microbial community composition. However, this can be achieved through structural equation modeling (SEM). Our SEM models found that different soil physicochemical properties directly and indirectly affected soil bacteria diversity, richness and composition (Figure 7). TN and pH showed a significant indirect effect on the diversity of the soil bacterial community. TN indirectly affected soil bacterial diversity by impacting NO_3_^−^N content, while NH_4_^+^N, NO_3_^−^N, and SOC showed a significant direct effect on soil bacterial diversity. Additionally, we also observed SOC and NO_3_^−^N negatively correlated with soil bacteria diversity, whereas NH_4_^+^N was positively correlated with soil bacterial diversity. Soil bacterial richness serves as a crucial indicator for assessing the diversity of bacterial communities. Our study found that the content of SOC and NH_4_^+^N directly impacted the soil bacterial richness, while TN indirectly impacted the soil bacterial richness by affecting the content of NH_4_^+^N. SOC has a negative effect on soil bacterial richness, while NH_4_^+^N has a positive effect on soil bacterial richness. Our result is consistent with previous studies [91]. In addition, pH can also indirectly affect the soil bacterial diversity by changing the content of SOC. This is because the pH is usually negatively correlated with SOC content, which aligns with prior studies; for example, Lu et al. [92] analyzed 6102 bacterial samples from China, and found a negative correlation between soil pH and SOC concentration. Therefore, we supposed that soil pH serves as a major predictor of microbial diversity.

In the context of soil bacterial composition, our findings reveal that both total nitrogen (TN) and pH have direct effects on the soil bacterial community composition. SEM revealed that soil pH obviously correlated with soil microbial composition and that a path coefficient of soil pH was 2.699 in SEM. Moreover, the pH can indirectly mold community composition by exerting its influence on TN levels (Figure 7). These outcomes underscore the central role of pH as the primary driver in shaping soil attributes at the structural level. These observed results can be attributed to a multitude of factors. Primarily, pH levels directly shape bacterial communities, with most bacteria demonstrating a relatively limited pH tolerance range. Once this range is exceeded, their growth experiences a pronounced setback, leading to a forfeiture of their competitive edge and eventual replacement by other bacterial counterparts. Previous research showed that just a 25% reduction in growth compared to optimal conditions is enough to rapidly trigger competition among bacteria, ultimately giving an advantage to those with unhindered growth [93]. This interplay highlights the intricate dynamics between pH and soil bacterial communities, underscoring their paramount importance within soil ecosystems [94].

The diversity and structure of soil bacterial communities are shaped by a combination of soil physical and chemical properties. However, the mechanisms of their influence remain complex and understudied in the current literature. Further research should prioritize a comprehensive exploration of the mechanisms governing both biotic–abiotic and biotic–biotic interactions in soil ecosystems. Identifying and understanding these mechanisms is crucial for advancing our knowledge of soil ecology and its implications for sustainable development and ecosystem management.

## 5. Conclusions

This study found that soil physicochemical properties and soil bacterial communities together changed significantly between Naoli River, Honghe and Qixing River National Nature Reserve in the Sanjiang Plain. The most relative abundance bacterial genera in all soil samples were *RB41*, *Haliangium*, *Candidatus_Solibacte* and *Ellin6067*, and changed significantly between the three nature reserves. The relative abundance of soil bacterial genera *Haliangium* and *Candidatus_Solibacter* were higher in the NLH, but the relative abundance of soil bacterial genera *RB41* and *Ellin6067* were higher in both NW and QXH than those of other soil bacterial genera. Soil pH was the key environmental factor affecting soil bacterial community structure. The study supplied a deeper insight into the distribution of soil bacterial communities in the Sanjiang Plain, and a good example to understand the complex relationship between soil physicochemical properties and the diversity and structure of soil bacterial communities, as well as further emphasizing the importance of soil ecosystem in protecting the Sanjiang plain wetland.

## Figures and Tables

**Figure 1 microorganisms-11-02950-f001:**
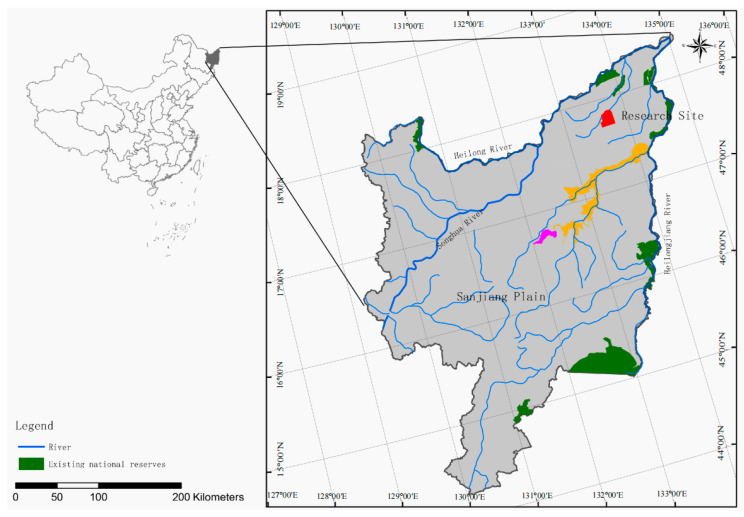
Locations of Honghe National Nature Reserve (red), Naoli River National Nature Reserve (orange) and Qixing River National Nature Reserve (pink) in the Sanjiang Plain, Heilongjiang Province, China.

**Figure 2 microorganisms-11-02950-f002:**
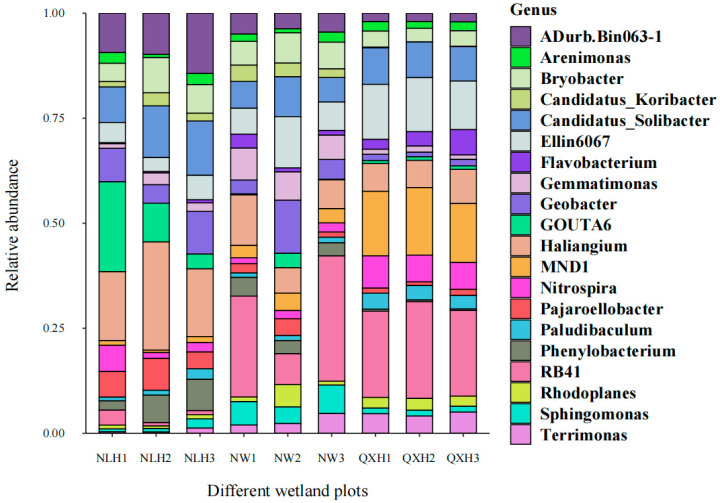
Effect of the different nature reserves on the relative abundance of soil bacterial genera. NLH, Naoli River National Nature Reserve; NW, Honghe National Nature Reserve; QXH, Qixing River National Nature Reserve.

**Figure 3 microorganisms-11-02950-f003:**
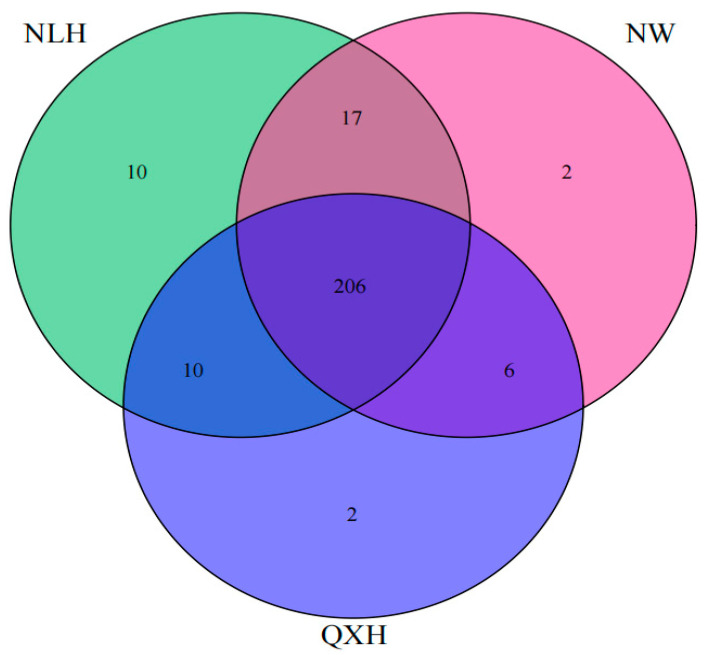
Venn diagram of shared and unique soil bacterial OTUs among three nature reserves. NLH, Naoli River National Nature Reserve; NW, Honghe National Nature Reserve; QXH, Qixing River National Nature Reserve.

**Figure 4 microorganisms-11-02950-f004:**
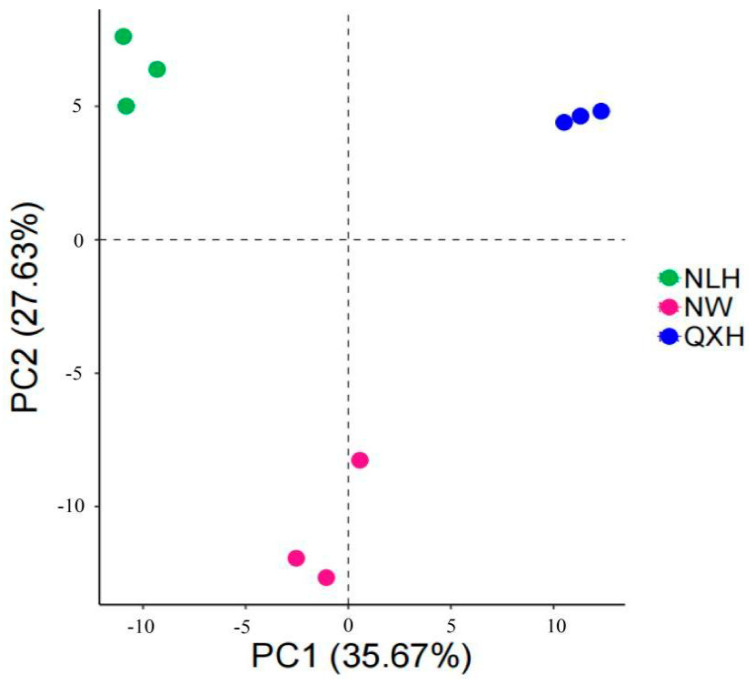
PCA analysis of the soil bacterial communities in the three nature reserves. NLH, Naoli River National Nature Reserve; NW, Honghe National Nature Reserve; QXH, Qixing River National Nature Reserve.

**Figure 5 microorganisms-11-02950-f005:**
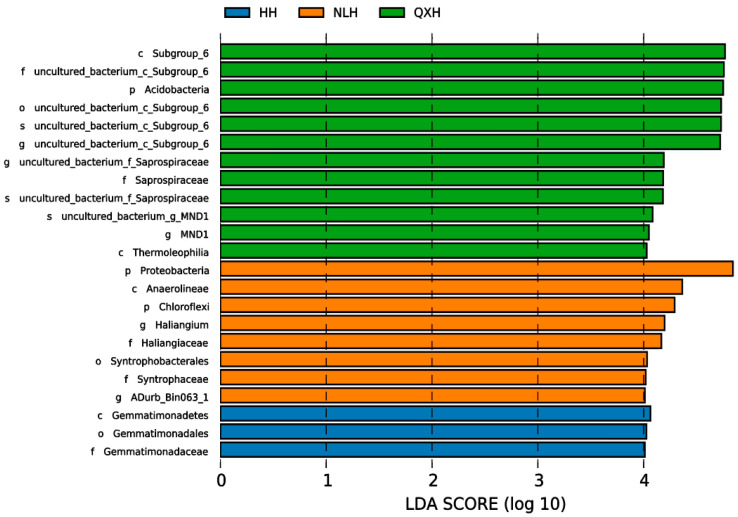
Lefse graph of the bacterial communities with LDA > 4.0 among different nature reserves. NLH, Naoli River National Nature Reserve; NW, Honghe National Nature Reserve; QXH, Qixing River National Nature Reserve.

**Figure 6 microorganisms-11-02950-f006:**
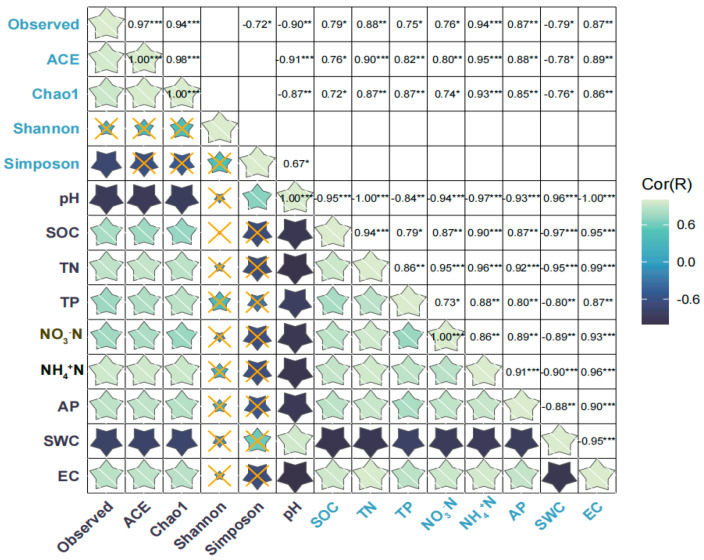
Heat map showing correlations between soil bacterial alpha diversity and measured targeted characteristics of the soil in the nature reserves. Statistical significance is given as * *p* < 0.05, ** *p* < 0.01, *** *p* < 0.001 and the "×" means not relevant. The color gradient represents the magnitude of the correlation between soil bacterial α-diversity and measured targeted characteristics of the soil in different nature reserves (lighter colors represent positive correlations and darker colors represent negative correlations). SOC: soil organic carbon; TN: total nitrogen; TP: total phosphorus; AP: available phosphorous; NH_4_^+^N: ammonia nitrogen; NO_3_^−^N: nitrate nitrogen; SWC: Soil Water Content; EC: electrical conductivity.

**Figure 7 microorganisms-11-02950-f007:**
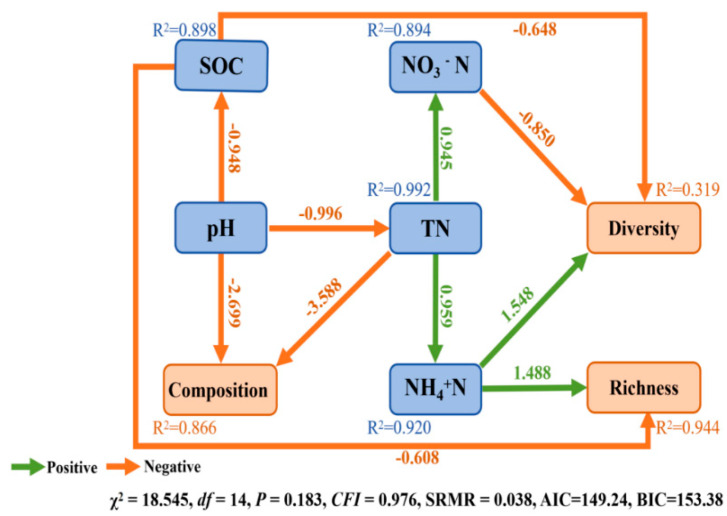
Path diagrams of structural equation modeling (SEM) of changes in measured targeted characteristics of the soil in relation to soil bacterial α-diversity (Shannon and richness indices) and composition (PCA1) of measured soils in different nature reserves. SOC: soil organic carbon; TN: total nitrogen; NH_4_^+^N: ammonia nitrogen; NO_3_^−^N: nitrate nitrogen; SWC: Soil water content. Values above the line represent the path coefficients. The orange lines indicate positive path coefficients and the green lines indicate negative path coefficients.

**Table 1 microorganisms-11-02950-t001:** Soil physicochemical properties of the three nature reserves.

Site	pH	SOC g/kg	TN g/kg	TP g/kg	AP mg/kg	NO_3_^−^N	NH_4_^+^N	SWC (%)	EC μs/cm
NLH	4.5 ± 0.2 b	53.9 ± 0.1 a	5.8 ± 0.2 a	1.0 ± 0.1 a	1.15 ± 0.1 a	23.7 ± 0.8 a	24.7 ± 0.8 a	42.21 ± 0.27 b	270.7 ± 1.4 a
NW	4.4 ± 0.1 b	54.7 ± 0.6 a	5.7 ± 0.1 a	0.99 ± 0.1 a	1.17 ± 0.1 a	23.8 ± 1.2 a	25.1 ± 0.1 a	42.27 ± 0.28 b	274.9 ± 3.2 a
QXH	7.3 ± 0.1 a	50.5 ± 1.0 b	2.9 ± 0.1 b	0.8 ± 0.1 b	1.0 ± 0.1 b	18.9 ± 0.5 b	21.7 ± 0.5 b	45.32 ± 0.91 a	194.8 ± 5.1 b

Values are given as mean ± standard error; different letters represent significant differences between treatments (*p* < 0.05). SOC: soil organic carbon; TN: total nitrogen; TP: total phosphorus; AP: available phosphorous; NH_4_^+^N: ammonia nitrogen; NO_3_^−^N: nitrate nitrogen; SWC: Soil Water Content; EC: electrical conductivity.

**Table 2 microorganisms-11-02950-t002:** Alpha diversity of the soil bacteria in the different nature reserves.

Site	OTU Richness			OTU Diversity
	Observed	ACE	Chao1	Shannon
NLH	1774.67 ± 84.44 a	1906.64 ± 57.37 a	1961.29 ± 60.06 a	9.36 ± 0.28 a
NW	1808.33 ± 7.59 a	1908.93 ± 15.96 a	1948.82 ± 31.04 a	9.16 ± 0.07 a
QXH	1622.67 ± 37.31 b	1757.77 ± 47.36 b	1818.84 ± 40.90 b	9.23 ± 0.04 a

Values are given as mean ± standard error; different letters represent significant differences between treatments (*p* < 0.05). The abbreviations are the same as those in Table 1.

## Data Availability

Data are contained within the article.

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
