# Peer review of "pH-Related Changes in Soil Bacterial Communities in the Sanjiang Plain, Northeast China"

_microorganisms, 2023, doi:10.3390/microorganisms11122950_

Round 1

Reviewer 1 Report

Comments and Suggestions for Authors

The study "Soil pH drive changes in soil bacterial communities in the Sanjiang Plain, Northeast China" link the changes in the microbial community composition of soil to nitrogen content.

Other studies showed that salinity was the major environmental determinant of microbial community composition than temperature, pH, and other physicochemical factors across global diverse environments (https://doi.org/10.1016/j.jare.2023.06.015,  10.1073/pnas.0611525104). The authors have to discuss these findings compared to their results. 

Did you measure the salinity among the physicochemical properties of the soil samples

Other studies identified Streptomyces spp. in soil samples (10.3390/microorganisms9061131), have you identified Streptomyces spp. in your study.

The authors have to check the format of the references throughout the manuscript to match the MDPI, also revise the names of microbial species.

Comments on the Quality of English Language

Moderate editing of English language required

Author Response

Dear Editor Sylvia Shi,

Dear Reviewer 1,

Thank you very much for giving us the opportunity to revise our manuscript entitled “Soil pH drive changes in soil bacterial communities in the Sanjiang Plain, Northeast China(microorganisms-2709022)”. Thank you very much for your valuable comments on our MS. According to your comments and suggestions, we carefully revised throughout our MS. Our detailed letter of response is contained in the attached document.

Reviewer 2 Report

Comments and Suggestions for Authors

Dear editor,

Thanks  for inviting me to review this interesting paper. Attached, you can find many comments that I hope can help the authors to improve the paper. In my opinion, the main problem is the absence of a clear explanation of the novelty of the paper. Now it seems a simple study case, and the novelty is the wetland… remark this. Photos of the study area would be great. Also, the authors have to explain if these conclusions can be obtained just with one sampling campaign once. Also, you measure many variables but the title only talks about pH. Looking forward to reading the next version.

Best regards

Comments on the Quality of English Language

Author Response

Dear Editor Sylvia Shi,

Dear Reviewer 2,

Thank you very much for giving us the opportunity to revise our manuscript entitled “Soil pH drive changes in soil bacterial communities in the Sanjiang Plain, Northeast China(microorganisms-2709022)”. Thank you very much for your valuable comments on our MS. According to your comments and suggestions, we carefully revised throughout our MS. Our detailed letter of response is contained in the attached document.

Reviewer 3 Report

Comments and Suggestions for Authors

1. Presentation of the article.

1.1. The article is formatted sloppily. Spaces between words are missing in many places. The legend below Table 2 preserves the undeleted words of the draft. Page 12, paragraph 2: the text of the article contains the text of the journal’s instructions (“Authors should discuss the results and how they can be interpreted from the pers”). Some abbreviations are given without explanation in the appropriate place.

1.2. English is not my native language. Nevertheless, I need to mention that text requires, at least, a moderate editing of English language. The beginning (Introduction, the first lines): “Wetlands are formed by the interaction of organisms, soil and hydrology. It has unique ecosystem structures” – i) hydrology is a science or scientific discipline, so, organisms can have interaction with water system of the ecosystem but not with a discipline. We cannot say “dogs interact with zoology”; ii) “wetlands” is plural, so, not “It has unique…”but “They have unique…”; etc.

2. Title.

Please change the title for more precise one: “pH-Related Changes in Soil Bacterial Communities in the Sanjiang Plain, Northeast China”.

3. Abstract.

3.1. “In this study, we assessed … to analyze the main driving factors.” This sentence produced an impression that authors checked dozens of factors and selected the main ones. Please list briefly in the abstract which factors were actually analyzed.

3.2. The abstract contains several abbreviations without explanation (TN; SOC) what they mean. Readers should get as much information as possible from the abstract without having to read the entire article to decode it. Please explain the abbreviations.

3.3. Please delete two last sentences of the abstract because they give readers no information at all. “Our results suggest that the soil physicochemical play a key role in shaping the community composition and diversity of soil bacteria”- it has already been known before. “This study will enable us to better understand how changes in soil physical and chemical properties drive changes in soil bacterial communities in the Sanjiang Plain” – all readers will be happy to know what is your better understanding. Please supplement the abstract with a specific example of shifts in bacterial community related with the pH change.

4. Key words.

Let us be more specific: i) remove “soil microorganisms” while you studied only soil bacteria, ii) replace unclear “soil physical and chemical properties“ for the measured ones: pH; nitrogen; phosphorous; organic matter.

5. Introduction.

It would be nice if the authors did not limit the text with some generalized informayion. Please supplement the introduction with some list of abiotic factors including temperature, humidity, redox and oxygenation of the soil and then add some short explanation about your work hypothesis: why you have selected only pH, nitrogen, phosphorus and total organic carbon as the main factors.

6. Materials and Methods.

6.1. Please change the very generalized sentence “The active cumulative temperature ranges from 2,300 to 2,500°C above 10°C, highlighting significant thermal accumulation“ for more specific: typical temperature for seasons varied from … to …; the study was performed in the … months when the soil temperature varied from … to …

6.2. Subsection “2.3. Characterization of Soil Physicochemical Parameters” is of very great significance for this article, so, please rewrite it more detailed.

6.2.1. Please mention the devices you used (name, company of production, country). As well, please, present the references (publications) for the detailed descriptions of the used methods.

6.2.2. Too short presentation of this subsection provokes a lot of questions:

- what was the temperature of soil during the sampling, how did you measure it, what was the difference in temperature between different spots during the study?

- “to measure the soil moisture content of each plot by gravimetric method” – where are the data on the soil humidity (water content) for the studied sites? Is the abbreviation SWC for the soil water content?

- “Soil pH was measured by pH tester after shaking the soil-water (1:2.5 w/v) suspension for 30 minutes” - what do you think about possible loss of CO2 from the samples during 30 min shaking and, thus, shifts of pH? did you check stability of pH?

- “soil effective phosphorus by molybdenum-antimony antimicrobial colorimetry”- what do you mean with “antimicrobial colorimetry”?

- “soil organic carbon by potassium dichromate oxidation-spectrophotometry” – potassium dichromate reacts not only with organic compounds but with reduced inorganic compounds too. So, it would be nice to present some data on redox conditions or on oxygen concentration in the watered soil, etc.

6.3. Subsection “2.5. Analysis of Sequencing Data”. Why did you prefer to use FUNGAL database for classification of BACTERIA?

7. Results & Discussion sections.

7.1. Please explain abbreviations and “different letters” (as you named them) given in the Table 1.

7.2. Figure 1 mentioned “the relative abundance of soil bacterial genus” (more correct not genus but plural “genera”). However, the list in the Figure shows not only generic names but some other operational taxonomic units (OTUs). Do they also have the taxonomic level of genus? Please comment this state.

7.3. List of the genera names in the Figure 1 shows an absolute domination of Gram-negative bacteria. There are no spore-forming bacilli. It is very unusual for numerous soils. You need to comment this discovery.

7.4. Figure 5. Please change “soil physicochemical properties” for more precise definition, namely: “measured targeted characteristics of the soil”. This comment can be applied to some paragraphs of the text too.   

8. Conclusions.

Please complete the section with more specific data, for example: I) the analysis revealed the presence and dominance of 20 OTUs (genus level); II) the coincidence of bacterial communities in the 3 studied areas during the study period reached about ...%; a minority of mismatched groups were represented by... genera; iii) according to the measured soil characteristics, the main factors determining the composition and abundance of bacterial communities were related to the availability of nitrogen compounds.

Comments on the Quality of English Language

English is not my native language. Nevertheless, I need to mention that text requires, at least, a moderate editing of English language. The beginning (Introduction, the first lines): “Wetlands are formed by the interaction of organisms, soil and hydrology. It has unique ecosystem structures” – i) hydrology is a science or scientific discipline, so, organisms can have interaction with water system of the ecosystem but not with a discipline. We cannot say “dogs interact with zoology”; ii) “wetlands” is plural, so, not “It has unique…”but “They have unique…”; etc.

Author Response

Dear Editor Sylvia Shi,

Dear Reviewer 3,

Thank you very much for giving us the opportunity to revise our manuscript entitled “Soil pH drive changes in soil bacterial communities in the Sanjiang Plain, Northeast China(microorganisms-2709022)”. Thank you very much for your valuable comments on our MS. According to your comments and suggestions, we carefully revised throughout our MS. Our detailed letter of response is contained in the attached document.

Round 2

Reviewer 3 Report

Comments and Suggestions for Authors

Thank you for the thorough revision of the text.